# Uniaxial stress flips the natural quantization axis of a quantum dot for integrated quantum photonics

Xueyong Yuan[1], Fritz Weyhausen-Brinkmann[2], Javier Martín-Sánchez[1,3], Giovanni Piredda[4], Vlastimil Křápek[5], Yongheng Huo [1,6,7], Huiying Huang[1], Christian Schimpf[1], Oliver G. Schmidt[6], Johannes Edlinger[4], Gabriel Bester[2], Rinaldo Trotta[1,8] & Armando Rastelli[1]

The optical selection rules in epitaxial quantum dots are strongly influenced by the orientation of their natural quantization axis, which is usually parallel to the growth direction. This configuration is well suited for vertically emitting devices, but not for planar photonic circuits because of the poorly controlled orientation of the transition dipoles in the growth plane. Here we show that the quantization axis of gallium arsenide dots can be flipped into the growth plane via moderate in-plane uniaxial stress. By using piezoelectric strain-actuators featuring strain amplification, we study the evolution of the selection rules and excitonic fine structure in a regime, in which quantum confinement can be regarded as a perturbation compared to strain in determining the symmetry-properties of the system. The experimental and computational results suggest that uniaxial stress may be the right tool to obtain quantum-light sources with ideally oriented transition dipoles and enhanced oscillator strengths for integrated quantum photonics.

[1] Institute of Semiconductor and Solid State Physics, Johannes Kepler University Linz, Altenbergerstraße 69, 4040 Linz, Austria. [2] Institut für Physikalische Chemie, Universität Hamburg, Grindelallee 117, 20146 Hamburg, Germany. [3] Departamento de Física, Universidad de Oviedo, 33007 Oviedo, Spain. [4] Forschungszentrum Mikrotechnik, FH Vorarlberg, Hochschulstraße 1, 6850 Dornbirn, Austria. [5] Central European Institute of Technology, Brno University of Technology, Purkyňova 123, 61200 Brno, Czech Republic. [6] Institute for Integrative Nanosciences, IFW Dresden, Helmholtzstraße 20, 01069 Dresden, Germany. [7] Hefei National Laboratory for Physical Sciences at Microscale, University of Science and Technology of China, Shanghai Branch, Xiupu Road 99, 201315 Shanghai, China. [8] Department of Physics, Sapienza University of Rome, Piazzale Aldo Moro 5, 00185 Rome, Italy. Correspondence and requests for materials should be addressed to A.R. (email: armando.rastelli@jku.at)

Semiconductor quantum dots (QDs) obtained by epitaxial growth are regarded as one of the most promising solid-state sources of triggered single and entangled photons for applications in emerging quantum technologies[1–4]. For this purpose, the radiative recombination of excitons consisting of electrons and holes occupying the lowest energy states confined in the conduction bands (CBs) and valence bands (VBs) of the semiconductor are employed. The selection rules for such transitions in QDs made of common direct-bandgap semiconductors are governed by the nature of the confined holes' ground-state (HGS). In turn, the HGS depends on the confinement potential defined by the structural properties of the QD and surrounding barrier. Epitaxial QDs usually possess flat morphologies, and heights smaller than the Bohr radius of the confined excitons. Carriers are therefore strongly confined along the growth ($z$) direction, which also defines the natural quantization axis for the total angular-momentum operator for the VB states[3–5]. The vertical confinement splits the heavy-hole (HH) and light-hole (LH) bands, so that the HGS has dominant $HH_z$ character, with total angular-momentum projection $J_z = \pm 3/2$ (in units of $\hbar$). Dipole-allowed transitions involving such states are characterized by transition dipoles perpendicular to $z$, making them well suited for efficient vertically emitting single-photon devices[6–11]. For planar integrated quantum photonics applications[4,12–18], it would be instead desirable to have QDs with transition dipoles perpendicular to the propagation direction, and hence a quantization axis with well-defined orientation in the $x$–$y$ plane. In fact, the azimuthal orientation of the transition dipoles for $HH_z$-excitons is usually affected by random fluctuations[19,20], preventing their optimal coupling to guided modes.

In spite of their importance, very little effort has been devoted to developing quantum sources optimized for planar photonic circuits. An exception is represented by ref.[21], where nanowires-QDs were removed from the substrate and turned by 90° for efficient coupling to dielectric waveguides. Here we show how the natural quantization axis of a QD can be turned to lie in the growth plane without actually rotating the semiconductor matrix, thus preserving the compatibility of the QD heterostructure with planar photonic processing. To this aim, we use two key ingredients: (1) high-quality, initially unstrained GaAs QDs[22–24] with a relatively large height, see Fig. 1a; (2) in-plane uniaxial stress, provided by micro-machined piezoelectric actuators featuring geometric strain amplification. Different from most previous experiments, in which stress was added after growth as a perturbation to fine tune the emission properties of QDs[25,26], confinement can be seen here as a perturbation compared to the strain-induced effects in determining the orientation of the quantization axis and hence the optical selection rules and excitonic fine structure.

## Results

**Illustration of the concept.** As mentioned above, the HGS of epitaxial QDs has dominant $HH_z$ character as a consequence of the vertical confinement. The same situation is encountered when biaxial compression in the $x$–$y$ plane is applied to bulk GaAs. In these cases, the angular dependence of the $HH_z$ Bloch-wavefunction shows a "donut" shape (see inset in Fig. 1b). Combined with the s-like electron Bloch-wavefunction, such a state couples only to light with polarization perpendicular to the $z$ quantization axis.

Obviously, for bulk GaAs, the quantization axis could be set to the [100] ($x$) direction by simply applying a biaxial stress in the $y$–$z$ plane or uniaxial stress along the $x$-direction. Now the question is: is it possible to obtain a QD with an in-plane quantization axis through the application of realistic stress values? Previous

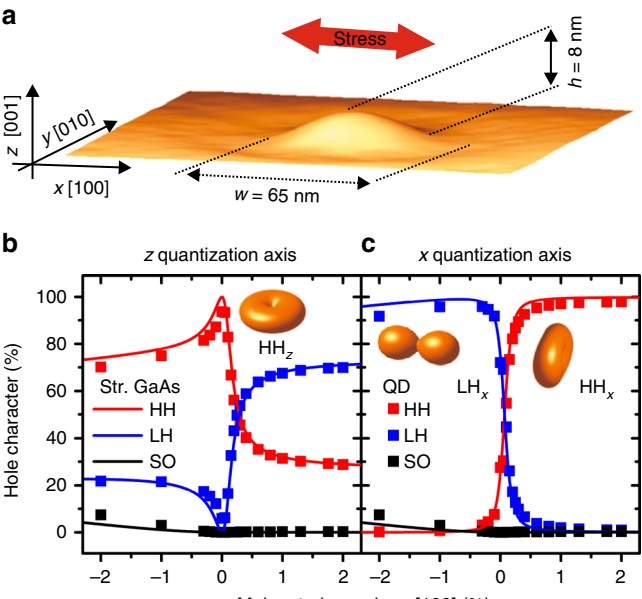

**Fig. 1** Illustration of the concept used to rotate the natural quantization axis of an epitaxial QD. **a** 3D view of an AFM image of a GaAs QD embedded in AlGaAs matrix. **b, c** Calculated effect of uniaxial stress along the [100] crystal direction on the degree of mixing of the topmost VB using the $z$ and $x$ quantization axis, for bulk GaAs subject to a fixed in-plane biaxial compression with $\sigma_{xx} = \sigma_{yy} = -120$ MPa (solid curves) and for the experimentally studied QDs (symbols). The plots illustrate the importance of the choice of the quantization axis when discussing VB mixing and show that the natural quantization axis of the chosen QDs can be flipped with moderate strains. Insets: Angular dependence of the probability density distribution of the Bloch wavefunctions of the topmost VB states at the $\Gamma$ point showing the conversion of a $HH_z$ state into a $HH_x$ ($LH_x$) state under tension (compression)

experiments have shown that symmetry breaking in the $x$–$y$ plane results in substantial $HH_z$–$LH_z$ hole-mixing[27], but the possibility of reaching pure $HH_x$ or $LH_x$ states has not been discussed so far. And in fact the answer would be negative for conventional Stranski–Krastanow QDs because vertical confinement and in-plane compression (of the order of GPa) "team up" to stabilize a vertically oriented quantization axis. Initially unstrained GaAs QDs with relatively low confinement energies are instead ideally suited to address the question. To quantify to what extent the quantization axis is oriented along the original $z$ direction or the desired $x$ direction under uniaxial stress, we calculated the HGS of our QDs (Fig. 1a) via the empirical pseudopotential method (EPM) and projected it onto the $HH_n$, $LH_n$, and $SO_n$ (split-off) states, i.e., the eigenstates of the angular-momentum-projection operator $J_n = \mathbf{J} \cdot \mathbf{n}$ along the quantization axis direction specified by the unit vector $\mathbf{n}$. The results for $\mathbf{n}$ parallel to the $z$- ($x$-) directions are shown with symbols in Fig. 1b and c. Using the conventional quantization axis $z$ (Fig. 1b), we would reach the wrong conclusion that uniaxial stress results in strong HH–LH mixing even for large strains. By using instead the new quantization axis $x$, we see that the topmost VB has almost pure $HH_x$ character (with some $LH_x$ admixture) upon sufficiently strong tension and almost pure $LH_x$ character (with some $SO_x$ admixture) upon compression (Fig. 1c). It is important to note that for intermediate values of strain the overlap of the HGS with the eigenstates of $J_n$ is rather poor for any $\mathbf{n}$, indicating that the HGS has low symmetry and a "good" quantization axis cannot be defined. This also means that uniaxial stress along $x$ produces a

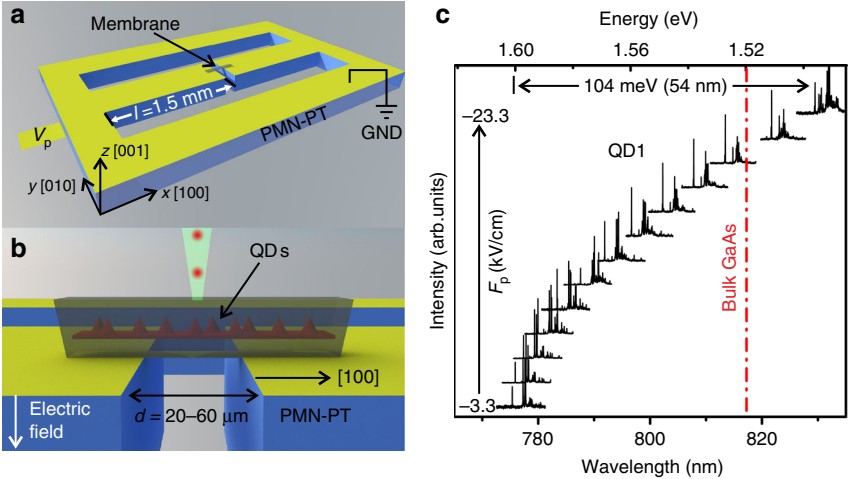

**Fig. 2** Experimental configuration and emission energy tuning of a GaAs QD via uniaxial stress provided by a micro-machined PMN-PT actuator. **a** Sketch of the actuator featuring two fingers with length $l$ separated by a gap of width $d$. A semiconductor membrane with embedded QDs is bonded on the fingers and forms a bridge above the gap. $V_p$ is the voltage applied to the bottom of the PMN-PT actuator with respect to the top contact, which is grounded. Because of the chosen poling direction, a negative electric field $F_p$ across the PMN-PT induces a contraction of the fingers and uniaxial tensile stress in the semiconductor along the $x$ direction. The coordinate system is the same as for the (Al)GaAs crystal. **b** Side-view of the device. PL measurements are performed by exciting and collecting PL along the $z$ axis. **c** Normalized PL spectra of a GaAs QD measured for increasing uniaxial tensile stress (from bottom to top). The large tuning range leads to emission below the bandgap of unstrained bulk GaAs (dashed line)

quantization-axis flip rather than a smooth rotation. (For more details, see Supplementary Notes 8–10 and Supplementary Movie.) Remarkably our EPM calculations predict that the "swapping" of quantization axis occurs already at moderate strains: the HGS of our QDs should have >90% $HH_x$ character for strains $\varepsilon_{xx} \gtrsim 0.3\%$ and >90% $LH_x$ character for $\varepsilon_{xx} \lesssim -0.1\%$. This result crucially relies on the use of tall and initially unstrained QDs and could not be achieved with conventional Stranski–Krastanow QDs (confirmed by experiments not shown here). The predicted evolution of the HGS upon uniaxial stress is robust and can be even caught with a simple model of bulk GaAs subject to a fixed biaxial stress in the $x$–$y$ plane – which mimics the symmetry-reducing effect produced by the vertical confinement – and variable uniaxial stress along the $x$ direction, see curves in Fig. 1b, c. Under tension, we expect a donut-shaped Bloch-wavefunction (right inset in Fig. 1c), a configuration suitable for light-coupling into $x$-oriented waveguides designed to sustain TE-like or TM-like modes (the $HH_x$-exciton emission is dominated by a $y$- and a $z$-oriented dipole). Under compression the wavefunction has instead a dumbbell shape elongated along the $x$-direction (left inset in Fig. 1c). This configuration is well suited for coupling into $y$-oriented waveguides designed to sustain TE-like modes (the $LH_x$-exciton emission is dominated by an $x$-oriented dipole).

Before discussing the future integration of stress-engineered QDs into quantum-photonic circuits, we present below the experimental proof of the concept.

**GaAs QDs under uniaxial tensile stress**. To test experimentally the above concept and to follow the evolution of the emission of single QDs under uniaxial stress, we have developed a piezo-electric actuator capable of "stretching" an overlying semiconductor layer up to its mechanical fracture. The actuator is made of $[Pb(Mg_{1/3}Nb_{2/3})O_3]_{0.72}[PbTiO_3]_{0.28}$ (PMN-PT) and combines the advantages of a strain-amplifying suspension platform[28,29] and continuously variable stress[26,30]. PMN-PT substrates were micro-machined to feature two fingers separated by a narrow gap, see Fig. 2a. A negative voltage $V_p$ applied to electrodes placed at the bottom of the two fingers leads to their

in-plane contraction and consequent gap expansion. While the maximum strain achievable in PMN-PT is limited to about 0.2% at low temperatures[26], strain in a layer suspended between the fingers (here a 250-nm-thick (Al)GaAs membrane containing GaAs QDs) is amplified by a factor $2l/d$, where $l$ is the length of each finger along the $x$-axis (here 1.5 mm) and $d$ is the gap width (20–60 μm). Our simple actuator is thus capable of delivering strain values comparable to state-of-the-art microelectromechanical systems[31] and provides a compact alternative to the commonly used bending method[32–34] for operation at cryogenic temperatures (here ~8 K for all measurements) in a cold-finger cryostat.

Strain amplification allows the emission energy of a QD to be continuously shifted in a spectral range exceeding 100 meV, as illustrated by the photoluminescence (PL) spectra of Fig. 2c. This shift, which is the largest achieved so far for QDs integrated on piezoelectric actuators, overcompensates the confinement energies and leads to emission well below the bandgap of bulk GaAs (see dashed line in Fig. 2c). From the comparison between the observed shift and the one calculated with EPM and the configuration interaction (CI) method, we deduce a maximum stress of ~1.3 GPa (strain ~1.5%), a value which is limited by mechanical fracture of the membrane.

We now focus on the experimental proof that uniaxial tensile stress along the [100] ($x$) direction leads to a HGS with $HH_x$ character, manifesting in characteristic optical selection rules. To this aim, we performed linear polarization-resolved measurements of the PL of various GaAs QDs under increasing stress while collecting light emitted along $z$, see sketch in Fig. 2b. Figure 3a shows color-coded PL spectra of an almost unstrained QD embedded in a large (~2×4 mm²) membrane. (A small positive electric field $F_p$ was applied to the actuator to partially compensate for some residual processing- and cooling-induced stress). The neutral exciton (X) emission shows a wavy pattern as a function of polarization direction $\varphi$ due to the "fine-structure splitting"[35] (FSS) caused by slight in-plane anisotropy of the confinement potential[36,37]. Several multiexcitonic (MX) lines are also observed, which stem from recombination of a ground-state electron with a HGS (as for X), but in presence of additional photo-generated carriers in the QD. Overall, the initial spectra

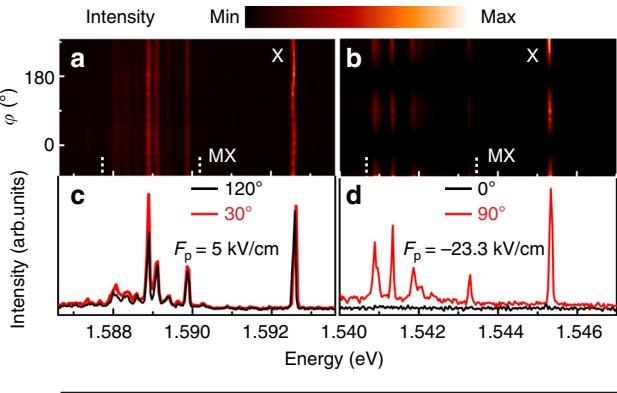

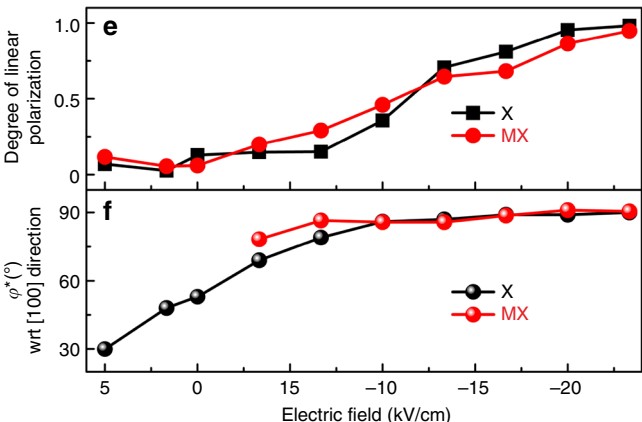

**Fig. 3** Rotation of natural quantization axis under uniaxial stress along [100]. **a–d** Color-coded linear polarization-resolved PL spectra of a GaAs QD for increasing tensile stress (**a**, **b**) and selected spectra along orthogonal polarization directions (**c**, **d**). The polarization angle $\varphi$ is referred to the [100] crystal direction. Initially, **a**, **c** the emission from the neutral exciton X and multiexcitons MX shows no significant net polarization and the X emission is characterized by two bright components linearly polarized along orthogonal but random directions (~30° and ~120° for this QD). This random orientation stems from slight anisotropy in the confinement potential defined by the QD and also from some process-induced prestress. At large stress, **b**, **d** the PL is almost fully polarized along the [010] direction ($\varphi^{\star} = 90°$). **e** Evolution of the degree of linear polarization of X emission and MX "band" (emission between vertical dashed lines in **a** and **b** for increasing tensile stress (increasing magnitude of applied electric field on actuator). **f** Evolution of polarization orientation $\varphi^{\star}$ for the MX band and for the high-energy component of the X line for varying stress

(Fig. 3a, c) show no net polarization in the $x$–$y$ plane, indicating weak $\mathrm{HH}_z$–$\mathrm{LH}_z$ mixing[27,38,39]. The scenario changes completely under strong tension, as shown in Fig. 3b and d for the same QD: Light from X and MX becomes fully polarized parallel to the $y$-direction. This is exactly what we expect for a $\mathrm{HH}_x$-exciton, since the limited numerical aperture of the used objective (0.42) prevents us collecting $z$-polarized light. The degree of linear polarization, defined as $P = (I_{\max} - I_{\min})/(I_{\max} + I_{\min})$, with $I(\varphi)$ being the integrated intensity for the X or the MX lines, is shown in Fig. 3e. $P$ increases with increasing stress and the angle $\varphi^{\star}$ for which $I(\varphi^{\star}) = I_{\max}$ aligns with the $y$-direction (perpendicular to the $x$ quantization axis for the $\mathrm{HH}_x$ state), see Fig. 3f. Because of the competition between intrinsic VB-coupling induced by the low symmetry of the nanostructure and the strain-induced effects, the transition from a $\mathrm{HH}_z$ to a $\mathrm{HH}_x$ is smooth, as already anticipated in Fig. 1b and c. However, for sufficiently large stress, the in-plane optical dipoles of X and MX transitions are deterministically aligned perpendicular to the pulling direction,

independent of the initial orientation, as shown in Fig. 3f and Supplementary Notes 4, 6.

The picture is not yet complete: an $\mathrm{HH}_x$–exciton couples not only to $y$-polarized but also to $z$-polarized light. To observe such a signature without resorting to side collection[27,40] and to study in detail the evolution of the X fine structure, we have used narrow membranes (~3.5 μm width) with tilted edges (obtained by wet chemical etching, see Supplementary Note 5), which we expect to partially deflect $z$-polarized light into the collection path (see sketch in Fig. 4a). After collecting and averaging polarization-resolved PL spectra of a QD for fixed values of the voltage applied to the actuator, we plot the averaged spectra using one of the bright exciton lines (indicated with $B_{y'} \to B_y$ in the following) as a reference for the energy axis. The result for a representative QD is shown in Fig. 4b. (We have measured in detail the behavior of two additional QDs finding fully consistent results, see Supplementary Note 6). Initially, the emission is characterized by two lines ($B_{x',y'}$), which are linearly polarized perpendicular to each other in the $x$–$y$ plane and are split by the FSS, see Fig. 4c. This is the typical signature of a $\mathrm{HH}_z$ exciton. For the investigated QDs, the orientations $x'$ and $y'$ is rather random (see, e.g., Fig. 3a) and the average FSS is ~4 μeV before processing[20]. We stress again that fluctuations in the azimuthal orientation of the transition dipoles are commonly encountered also for other QDs[19] and represent a problem if light has to be efficiently coupled into a specific mode of a waveguide. With increasing stress, the energy separation between the two lines increases, the polarization direction of the high(low) energy component rotates and aligns to the $y(x)$ direction independent of the initial orientation (see also Fig. 3f), and the intensity of the low energy component drops monotonically. The latter observation indicates that $B_{x'}$ converts to a dark state ($D_x$). We note that in general $B_{x',y'}$ undergo an anticrossing at moderate strain levels, as long as the stress orientation does not coincide with the anisotropy axis defined by the confinement[26]. However, different from the behavior in the perturbative regime, in which the "FSS" varies linearly with the applied stress[41] away from the anticrossing, the $(B_{y'} \to B_y) - (B_{x'} \to D_x)$ splitting shows a sublinear increase for all investigated QDs.

The most striking feature emerging from Fig. 4b is an additional line appearing first as a shoulder at the low energy side of $B_{y'} \to B_y$ ($D_z$ in Fig. 4d) and then moving to its high-energy side with increasing stress. Since such a component appears as $y$-polarized and was not observed in any QDs embedded in large membranes, we can safely attribute it to an initially dark exciton $D_z$, which evolves into a $z$-polarized bright exciton $B_z$. The final configuration at large stress (Fig. 4e) is fully consistent with a $\mathrm{HH}_x$ HGS: two bright excitons ($B_y$ and $B_z$), which have polarization perpendicular to the new quantization axis $x$ and two dark excitons. In the ideal case of pure $\mathrm{HH}_x$ states, the Bloch wavefunctions of such dark states (one of which remains dark throughout the experiment) are a linear combination of the states with $J_x = \pm 2$. The large "FSS" ($B_y - B_z$ splitting) can be qualitatively attributed to the strong anisotropy of the QD confinement potential in the $y$–$z$ plane (see Fig. 1a), leading to a low(high)-energy component $B_y(B_z)$ parallel to the long(short) axis of the QD, compatible with results obtained with $\mathrm{HH}_z$-excitons confined in GaAs QDs with elongation in the $x$–$y$ plane[37]. We note that the achieved configuration is markedly different from that of $\mathrm{LH}_z$-excitons obtained either in vertically elongated InGaAs nanostructures[42] or in GaAs QDs under in-plane biaxial tension[23]. In that case, the quantization axis is still along the $z$-direction and there are three bright optical dipoles (instead of two for the $\mathrm{HH}_x$ case): one is aligned along $z$ and the other two lie, randomly oriented, in the $x$–$y$ plane.

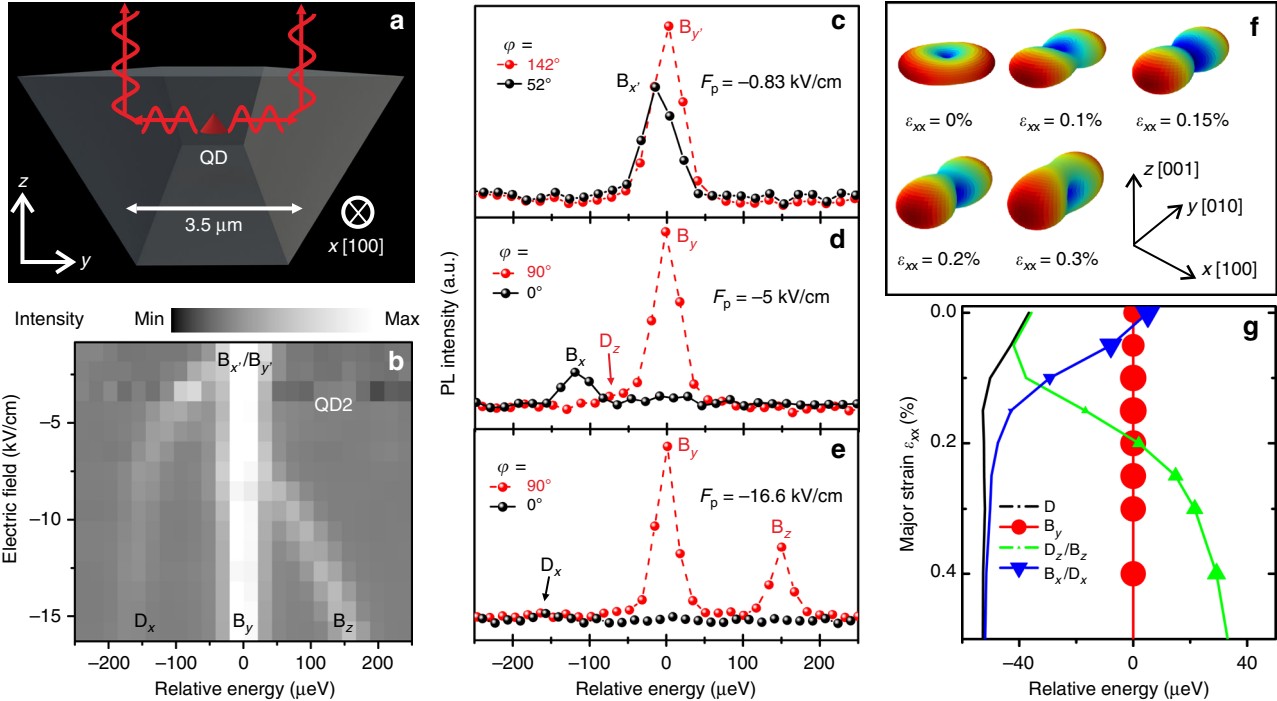

**Fig. 4** Evolution of the fine structure of a neutral exciton confined in a GaAs QD for increasing uniaxial stress and comparison with theory. **a** Sketch of the cross-section (y–z plane) of the stripe-like membrane used to project the vertically polarized component $B_z$ into the collection optics (see Fig. 2b). The pulling (x) direction is orthogonal to the y–z plane. From the geometry, we expect such light to appear as y-polarized. **b** PL spectra of the neutral exciton (X) emission of a QD as a function of electric field applied to the piezoelectric actuator. Spectra are shifted horizontally using the line labeled as $B_{y'}$ or $B_y$ as reference. **c–e** Representative linearly polarized PL spectra of the X emission in the same QD for different values of the field $F_p$ applied to the actuator (uniaxial stress). In **c**, we see the usual fine-structure splitting (FSS) of a $HH_z$ exciton, characterized by two orthogonally polarized lines. The observed polarization anisotropy, relatively large FSS (14 μeV), and random orientation of the polarization directions are mostly ascribed to $HH_z$–$LH_z$ mixing induced by QD anisotropy and process-induced prestress. The applied uniaxial stress aligns the transition dipoles along and perpendicular to the stress axis, as shown in **d** and **e**. z-polarized emission appears first as a shoulder on the low energy side of the $B_y$ line (marked as $D_z$ in **d**). The emission of a $HH_x$ exciton is shown in **e**. **f** Polar-coordinate representation of the excitonic transition dipole calculated by the EPM + CI. **g** Relative transitions energies of the four excitonic components under uniaxial stress (compressive, top and tensile, bottom) computed by EPM + CI. The symbol size is proportional to the oscillator strength of the transitions

Figure 4g shows the evolution of the X emission under tensile stress along [100] as calculated with EPM + CI. As in the experiment, energies are referred to the $B_y$ line. The size of the symbols is proportional to the strength of each transition. The calculations qualitatively reproduce all the experimentally observed features including the darkening of one of the initially bright excitons and the brightening of one of the initially dark excitons. Also the energy shifts follow the experiment, although the absolute magnitudes are systematically smaller—a discrepancy, which is currently under investigation. Finally, we note that the rotation of the quantization axis and the associated donut-shaped HH states from the z to the x direction is accomplished through a complex topological transformation of the HGS wavefunction, as reflected by the angular dependence of the electron–HGS transition dipole calculated by EPM + CI, see Fig. 4f. (Because the electron wavefunction is marginally affected by stress, such plots closely resemble the angular dependence of the Bloch wavefunctions shown in Fig. 1c and in the Supplementary Movie).

## Discussion

We conclude by discussing the possible implications of this work for integrated quantum photonics[4,12]. Particularly relevant features of quantum emitters for such applications are: (i) efficient light-coupling to guided modes; (ii) fast radiative recombination, which is beneficial for high photon rates and indistinguishability. The re-orientation of the quantization axis of a GaAs QD shown here addresses the first point. We stress that the used QDs have excellent optical properties in terms of single-photon purity and indistinguishability when excited resonantly through a 2-photon-absorption process[24,43,44], which is compatible with planar photonic circuits and strongly reduces exciton recapture[44,45]. The fact that no line broadening was observed up to the maximum achievable stress values (see, e.g., Fig. 4c–e) makes us confident that stress along [100] does not deteriorate such excellent properties.

Besides the possibility of orienting the transition dipoles perpendicular to the propagation axis, there are two additional appealing features of uniaxial stress engineering. First, under proper uniaxial tension (which may be achieved by suspending the QD source at the edge of a photonic chip followed by coating with a dielectric stressor[46]) it may be possible to achieve level degeneracy between the $B_y$ and $B_z$ lines and obtain a QD launching polarization entangled photon pairs in a photonic circuit using the biexciton—$HH_x$ exciton cascade. Second, if we inspect the calculated transition strengths of excitonic dipoles under uniaxial compression (shown in Fig. 5a but not yet explored experimentally), we see that the emission of a $LH_x$-exciton is dominated by a single emission line with polarization parallel to the compressive stress direction and with an oscillator strength which is almost twice that of each of the two bright

**a**

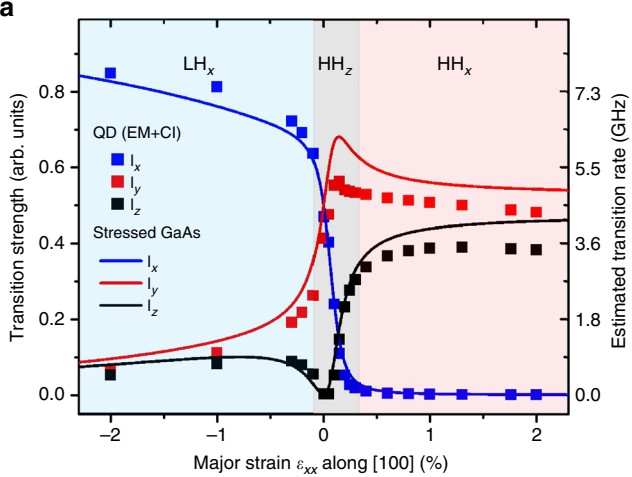

**b**

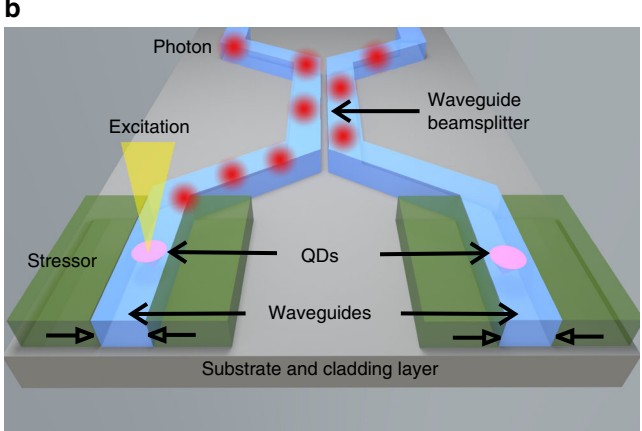

**Fig. 5** Envisioned applications of strain-engineered single-photon source for integrated quantum-photonic circuits. **a** Calculated transition strength and estimated rates for light polarization along the $x$ (blue), $y$ (red), and $z$ (black) directions for varying uniaxial stress along $x$ for QDs with structure taken from experiment (symbols) and GaAs subject to a fixed biaxial compression with $\sigma_{xx} = \sigma_{yy} = -120$ MPa (solid lines). **b** Envisioned approach to obtain high-speed single-photon sources based on $LH_x$ excitons confined in GaAs QDs. A waveguide is fabricated to contain preselected QDs followed by deposition of side dielectric layers, which act at the same time as cladding, passivation and stressor layers. The uniaxial stress (produced, e.g., by the different thermal expansion coefficients of dielectric and (Al)GaAs heterostructure aligns the quantization axis of the QD along the stress direction ($x$) and the resulting $LH_x$ exciton acts as an ultrafast source of single photons ideally matched to the propagating modes in the waveguide

excitons in our $HH_z$ QDs. Considering that $HH_z$ excitons in GaAs QDs are already characterized by very short lifetimes (~250 ps)[24,47], the combination of deterministic QD positioning in photonic structures[48,49] and uniaxial-strain engineering (see sketch in Fig. 5b) may lead to ideal single-photon sources with enhanced recombination rate (~8 GHz, see right axis of Fig. 5a) and indistinguishability levels. Instead of resorting to the Purcell effect[7,11] lifetime reduction would rely here on the strain-induced suppression of $y$- and $z$-polarized emission and consequent "oscillator-strength concentration" on the $x$-polarized emission. Even if technologically challenging, replacing the static stressors shown in Fig. 5b with piezoelectric materials would allow fine tuning the emission of different QDs to the same energy and to

achieve scalable photon sources for integrated quantum photonics circuits.

## Methods

**Computational methods.** For 3D calculations, we used the QD shape as directly provided by the digital form of a representative AFM image. We first relaxed the 6.5 million atomic positions, including the QD and a sufficiently large $Al_{0.4}Ga_{0.6}As$ alloy barrier using the valence force field approach[50] to minimize the strain energy. The determined positions are used as input into the atomistic empirical pseudo-potential method using the strained linear combination of bulk bands[51]. The ensuing single-particle wavefunctions and eigenenergies are subsequently used into a screened configuration interaction framework[52] where fully correlated exciton and multiexciton states are calculated. The required Coulomb and exchange integrals are microscopically screened according to the bulk model of Thomas and Fermi. The dipole moments are calculated based on the correlated excitonic wavefunctions[53]. A similar approach was used to compute the properties of light-hole ($LH_z$) excitons in ref.[23] but with unscreened Coulomb and exchange integrals.

The curves shown in Figs. 1b, c, 5a were obtained from the Luttinger–Kohn and Pikus–Bir (PB) Hamiltonians including the HH, LH, and the SO bands[54] by using the eigenstates of the PB Hamiltonian (at the $\Gamma$ point) corresponding to the topmost twofold degenerate VBs. The effect of vertical confinement was introduced phenomenologically by adding a fixed biaxial compression in the $x$-$y$ plane. The strain values displayed in the abscissa of Figs. 1b, c, 5a are those corresponding to a variable uniaxial stress ranging from $-2$ to $2$ GPa along [100]. Additional details and calculations are presented in the Supplementary Notes 8–12.

**Sample growth.** The used sample was grown by molecular epitaxy (MBE) on semi-insulating GaAs (001) substrate and consists of GaAs QDs placed at the center of a symmetric slab consisting of 30 nm $Al_{0.4}Ga_{0.6}As$, 90 nm $Al_{0.2}Ga_{0.8}As$ and 5 nm GaAs on top and below the QDs. This active structure was grown on a 100-nm-thick $Al_{0.75}Ga_{0.25}As$ etch-stop/sacrificial layer, which was removed via selective etching for membrane transfer. On the bottom $Al_{0.4}Ga_{0.6}As$ barrier, nanoholes were produced by the Al-droplet-etching method. During subsequent deposition of 1.6 nm GaAs and annealing, such nanoholes act as a mold cast to obtain GaAs QDs. For additional details, see Supplementary Note 1.

**Device fabrication.** 300-μm thick, (001)-oriented, PMN-PT substrates were cut by a micromachining system equipped with a femtosecond laser[55,56]. After cleaning, the top surface of the resulting actuator was homogeneously coated with a Cr/Au layer acting as top (electrically grounded) electrode. On the bottom surface only the two PMN-PT fingers were metallized leaving the surrounding frame uncoated, so that electric-field-induced deformation is limited to the fingers. In order to integrate the semiconductor structures on the micro-machined actuators we first used photolithography and wet chemical etching to define mesa structures with the desired lateral size and shape (in our experiment we used two structures, one is about ~2 × 4 mm² and another is 3.5 × 300 μm²). The sides of such mesas are tilted because of photoresist undercut during etching, see Supplementary Note 4. After lithography, the [100] crystal direction of the (Al)GaAs heterostructure was aligned parallel to the actuator fingers. Bonding of such layers onto the actuator was achieved via a flip-chip process using SU8 photoresist as adhesive bonding layer because of its high bond strength and low process temperature. A series of chemical etching steps were then performed to remove the original GaAs substrate and $Al_{0.75}Ga_{0.25}As$ layer, leaving membranes bonded onto the PMN-PT actuator. We note that while the actuator shows a fully predictable behavior in the tensile regime, poorly controlled membrane buckling prevents us to obtain reliable results in the compressive regime. For additional details, see Supplementary Notes 2–3.

**Optical characterization.** For optical measurements, the device was mounted on the cold-finger of a He-flow cryostat and cooled to ~8 K. A continuous wave laser (with wavelength of 532 nm) was used to achieve above-bandgap excitation of carriers. A 50× microscope objective with 0.42 numerical aperture was used on top of the sample to both focus the laser and collect the PL signal. After passing through a half-wave plate and a fixed linear polarizer, the PL signal was analyzed with a 750-mm focal length spectrometer equipped with a 1800 lines/mm ruled grating and a liquid-nitrogen cooled Si CCD with 20 μm pixel size. Piezo creep was observed during the measurements, especially at high electric fields. Although active stabilization could be used[26], the effect on polarization-resolved spectra was compensated here after data acquisition, as discussed in the Supplementary Note 7. The reproducibility of the presented results was thoroughly tested by repeating similar experiments of several QDs, see Supplementary Notes 4, 6.

**Data availability.** The data that support the findings of this work are available from the corresponding author upon request.

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

## Acknowledgements

This work was supported by the FWF (P 29603), the Linz Institute of Technology, the BMBF (Q.com-H, Contract No. 16KIS0108), the EU project HANAS (No. 601126210), AWS Austria Wirtschaftsservice (PRIZE Programme, Grant No. P1308457), and the European Research Council (ERC) under the European Union's Horizon 2020 Research and Innovation Programme (SPQRel, Grant Agreement No. 679183). X. Yuan was supported by China Scholarship Council (CSC, No. 201306090010). V. Křápek was supported by MEYS CR, Project (No. LQ1601). J. Martín-Sánchez acknowledges the Clarín Programme from the Government of the Principality of Asturias and the Marie Curie-COFUND actions PA-18-ACB17-29. We thank J. Claudon (CEA, Grenoble), S. Portalupi (University Stuttgart), and M.A. Dupertuis (EPFL, Lausanne) for fruitful discussions; M. Reindl, D. Huber, J. Wildmann for help with the processing and optical characterization, and A. Halilovic, A. Schwarz, S. Bräuer, U. Kainz, and E. Vorhauer for technical support.

## Author contributions

X.Y. fabricated the devices with the help of H.H., C.S., and J.M.S., and performed optical measurements with the help of R.T. G.P. processed the piezoelectric actuators with support of J.E. J.M.-S. tested the actuator concept. Y.H. performed preparatory work and grew the sample, with support of A.R. and O.G.S. AR performed **k·p** calculations and

interpreted the results with help of F.W.-B., V.K., and G.B. F.W.-B. performed atomistic pseudopotential calculations with support of G.B. A.R. and X.Y. wrote the manuscript with input from all the authors. A.R. conceived and coordinated the project.

## Additional information

**Competing interests:** The authors declare no competing interests.

