## [Peer Review File · Nature Communications]

Reviewers' comments:

Reviewer #1 (Remarks to the Author):

This paper reports an impressive series of experiments showing the complete control of the quantization axis of semiconductor quantum dots (QDs) by the use of uniaxial stress. This leads, among others, to the controlled switching of the polarization of the dipole moment of excitonic transitions, which may be beneficial for the application of QDs as single-photon sources. The experiments are convincing and well explained. The paper can be considered for publication in Nature Communications once the following minor shortcomings are addressed:

- The authors mention several times that a dipole polarization in the plane of the layers is not ideal for planar integrated circuits and that changing it to the growth plane would be better. It is not entirely clear why this is the case. Polarisation in the growth plane implies emission in both the transverse-electric (TE) and transverse-magnetic (TM) polarization. Simultaneous control of both polarizations is very challenging in integrated circuits, so I do not see why this should be an advantage, unless emission of entangled pairs from biexcitons is sought (which would be very difficult to use in an integrated circuit anyway). I believe that the control of the dipole moment is important anyway, but the authors should give more concrete and compelling arguments.
- The authors should more clearly indicate what differentiates this work from previous work from the same group, namely ref. 21 Huo et al.
- On page 4 the authors state that obtaining ground states with pure light-hole nature is not possible in Stranski-Krastanow QDs. This is not completely true. TM-polarised ground state emission and lasing was previously observed in so-called "columnar" QDs obtained by Stranski-Krastanow growth (Li et al., Appl. Phys. Lett. 95, 221116 (2009)). While this type of QD does not provide a level of control comparable to the one reported by the authors, it should be mentioned.
- On page 13 the reference to Fig. 1(d) should be replaced by Fig. 5(a)
- In the methods section, it is mentioned that the effect of piezo creep was compensated after data acquisition. It is not clear what this "compensation" is and the procedure used should be explained in detail (possibly in the Supplementary).
- Some parts of the Supplementary information (e.g. Section 4) are repetitions of text already present in the Methods section and can be eliminated.

Reviewer #2 (Remarks to the Author):

The central claim of the paper by A. Rastelli et. al is the ability to manipulate the direction of the quantization axis of a quantum dot via a piezo-induced stress.

After reading the manuscript, I am convinced that the authors have achieved the claimed effect. The results are new and certainly warrant publication in some form. However, I do not think that this paper meets publication criteria for Nature Communications for two reasons:

1) The quantization axis rotation appears to be an "unintended consequence" of frequency-tuning with stress, and it is not predictably controlled by stress. It appears that the degree of orientation control is random. Moreover, it seems that there is no way to predict the behavior of a given dot under stress without an individual characterization of each dot.

2) The authors list potential benefits of quantization axis rotations (see "discussion"), but they are yet to demonstrate any of such benefits experimentally. For instance, if the authors would show any evidence of better (or worse) light-waveguide coupling after axis reorientation, the impact of their work would be significantly stronger.

In addition, there is a minor remark.

It is not obvious to me that fast radiative recombination is beneficial for indistinguishability. I believe that there are several publications on quality of single photons with different confinement, and stronger confinement does not necessarily result in better indistinguishability and photon purity. One of the physical reasons may be a dot re-excitation, see Phys. Rev. Lett. 109, 163601, the other - stronger coupling between acoustic phonons and quantum dots.

Reviewer #3 (Remarks to the Author):

This manuscript shows that the quantization axis of GaAs quantum dots can be changed through application of uniaxial strain perpendicular to the growth direction. The quantum dot samples are strained in a piezoelectric actuator that is designed to provide a large, tunable tensile strain, and photoluminescence of single dots is used to observe changes in the emission line strengths, emission energies, and polarization properties that indicate this change in the quantization axis. The results match a theoretical model of the band structure of the semiconductor. The primary motivation for this work is to change the polarization axis of the QD transitions to be more appropriate for in-plane photon emission in photonic structures.

The ability to control the properties of quantum emitters through strain seems quite interesting, and this approach using a piezoelectric device that amplifies strain seems quite novel and provides ways to change the QD properties in ways perhaps not possible before. The idea to change the quantization axis and polarization axes is a novel idea that has potential uses. The paper also combines the experimental observations with theoretical and computation results, making the observations and physics more clear. I recommend it for publication in Nature Communications and think it will be an influential paper in the areas of quantum dots and solid state quantum systems in general.

My only criticism is that the language used to describe the wavefunction of electrons in the quantum dot and the quantization axis is confusing to me, and I suspect it may be confusing to others. The main point of confusion for me is what is meant by the quantization axis. When starting to read the paper, I thought it was perhaps referring to the direction in which quantum confinement plays the biggest role – which would ordinarily be the smallest dimension of the quantum dot. After reading more carefully, I came to the conclusion that the paper is about the quantization axis of the total angular momentum for the valence band, which is changed by the application of strain. I'm still not sure that I'm completely understanding this right, so it would be very helpful to clarify this idea. Along these lines, the manuscript discusses how the data in Fig. 1b and 1c show a change in the quantization axis for different strains, but this really isn't clear to me. How is the direction of the quantization axis determined by this data? I'm really not sure what it means to plot the hole character for the x and z quantization axes. Related to this are statements about the confinement effects being small compared to the strain effects. That almost sounds like the carriers might not be confined to the dot, but I'm assuming this means that confinement may only play a small role in determining the quantization axis of the angular momentum. I think that clarifying this issue will strengthen the paper significantly. I have a few minor questions below that would also be useful to address.

Minor points:

- a. On page 9 the manuscript states that there are "two bright excitons (B_y and B_z), which have polarization perpendicular to the new quantization axis x and two dark excitons." When I read this I first thought it was saying that the dark excitons did not have polarization perpendicular to the quantization axis, but now I think this just means that maybe the electron and hole have a total angular momentum of 2 instead of 1, as in dark excitons with the angular momentum quantized along the z-direction. Could you clarify this point?
- b. Is there any reason why compressive strain could not be applied in this structure to look at the LH_x exciton?

Response to referees

Reviewers' comments:

Reviewer #1 (Remarks to the Author):

This paper reports an impressive series of experiments showing the complete control of the quantization axis of semiconductor quantum dots (QDs) by the use of uniaxial stress. This leads, among others, to the controlled switching of the polarization of the dipole moment of excitonic transitions, which may be beneficial for the application of QDs as single-photon sources. The experiments are convincing and well explained. The paper can be considered for publication in Nature Communications once the following minor shortcomings are addressed:

We thank the reviewer for her/his positive assessment and for recommending publication in Nature Communications. Her/his constructive remarks helped us improving the clarity of our manuscript, as detailed below.

COMMENT 1)

- The authors mention several times that a dipole polarization in the plane of the layers is not ideal for planar integrated circuits and that changing it to the growth plane would be better. It is not entirely clear why this is the case.

REPLY 1)

Indeed what counts is the orientation of the transition dipoles \underline{d} with respect to the electric field \underline{E}_k corresponding to the guided mode \underline{k} in a photonic waveguide. The coupling efficiency is in fact proportional to $|\underline{d} \cdot \underline{E}_k|^2$ [Rev. Mod. Phys. 87, 347–400 (2015)]. For optimal coupling, the QDs should thus feature at least one transition dipole parallel to \underline{E}_k . However, in conventional QDs, (with heavy-hole ground state, described by a total angular momentum projection $J_z = \pm 3/2$ along the z-quantization axis, which coincides with the growth direction) the azimuthal orientation of the transition dipoles (in the growth plane) is affected by random fluctuations [see, e.g. S. Seidl et al. Phys. E 40, 2153 (2008) or Y.H. Huo et al. Appl. Phys. Lett. 102, 152105 (2013)]. By setting instead the quantization axis for the holes total angular momentum in the growth plane, we achieve a deterministic control of the orientation of the transition dipoles.

To make this point clear we have taken the following actions:

- We have added a short explanation in the abstract “...but not ideal for planar photonic circuits because of the poorly controlled orientation of the transition dipoles in the growth plane”
- We have modified the last sentence of the introduction on page 2: “For planar integrated quantum photonics applications,^{4,12–18} it would be instead desirable to have QDs with transition dipoles perpendicular to the propagation direction, and hence a quantization axis with well-defined orientation in the x-y plane. In fact the azimuthal orientation of the transition dipoles for HH₂- excitons are usually affected by random fluctuations,^{19,20} preventing optimal coupling to guided modes.”
- In the new Fig. 3d we provide additional data showing that the in-plane transition dipoles align perpendicularly to the main stress direction under tension. This result (and similar results shown in Fig. S4d and Fig. S7e for other dots) demonstrates that – through the application of uniaxial stress along a chosen direction – we gain deterministic control on the orientation of the transition dipoles. We believe that this clarification addresses also the comment 1) of Reviewer 2.

COMMENT 2)

Polarisation in the growth plane implies emission in both the transverse-electric (TE) and transverse-magnetic (TM) polarization. Simultaneous control of both polarizations is very challenging in integrated circuits, so I do not see why this should be an advantage, unless emission of entangled pairs from biexcitons is sought (which would be very difficult to use in an integrated circuit anyway). I believe that the control of the dipole moment is important anyway, but the authors should give more concrete and compelling arguments.

REPLY 2)

We thank the reviewer for this remark. We can imagine that ridge waveguides with square cross-section and isotropic cladding (e.g. suspended waveguides) would sustain both TM and TE modes, but we prefer not to go into such details in the main text, since we do not have pertinent data. Following this remark we have decided nevertheless to extend a paragraph on page 5, right after presenting the concept. The text now reads: *“Under tension we expect a donut-shaped Bloch wavefunction (right inset in Fig. 1c), a configuration suitable for light-coupling into x-oriented waveguides designed to sustain TE-like or TM-like modes (the HH_x -exciton emission is dominated by a y- and a z-oriented dipole). Under compression the wavefunction has instead a dumbbell shape elongated along the x-direction (left inset in Fig. 1c). This configuration is well suited for coupling into y-oriented waveguides designed to sustain TE-like modes (the LH_x -exciton emission is dominated by an x-oriented dipole).”*

COMMENT 3)

- The authors should more clearly indicate what differentiates this work from previous work from the same group, namely ref. 21 Huo et al.

REPLY 3)

In that work the quantization axis was not altered by the applied stress (biaxial, in the x-y plane). As a consequence, the in-plane orientation of the transition dipoles remained random. To make this point clear and address also the following remark we have added a sentence on page 10: *“We note that the achieved configuration is markedly different from that of LH_z -excitons obtained either in vertically elongated InGaAs nanostructures⁴² or in GaAs QDs under in-plane biaxial tension.²³ In that case the quantization axis is still along the z-direction and there are three optical dipoles (instead of two for the HH_x case): one is aligned along z and the other two lie, randomly oriented, in the x-y plane.”*

COMMENT 4)

- On page 4 the authors state that obtaining ground states with pure light-hole nature is not possible in Stranski-Krastanow QDs.

REPLY 4)

We claim that it would be difficult to obtain a Stranski-Krastanow QD with a quantization axis lying in the growth plane. To avoid misunderstandings, we have rephrased the question on page 4: *is it possible to obtain a QD with an in-plane quantization axis through the application of realistic values of stress?*

COMMENT 5)

This is not completely true. TM-polarised ground state emission and lasing was previously observed in so-called "columnar" QDs obtained by Stranski-Krastanow growth (Li et al., Appl. Phys. Lett. 95, 221116 (2009)). While this type of QD does not provide a level of control comparable to the one reported by the authors, it should be mentioned.

REPLY 5)

We think the comparison to the case of a LH exciton with z-oriented quantization axis is indeed instructive. We have therefore added the sentence quoted under our reply 3) and the suggested reference.

COMMENT 6)

- On page 13 the reference to Fig. 1(d) should be replaced by Fig. 5(a)

REPLY 6)

We acknowledge the mistake pointed by the referee and corrected it.

COMMENT 7)

- In the methods section, it is mentioned that the effect of piezo creep was compensated after data acquisition. It is not clear what this "compensation" is and the procedure used should be explained in detail (possibly in the Supplementary).

REPLY 7)

In supplementary information, we added a new Section 7, which provides details on the data correction used to compensate the creep effect from the piezoelectric actuator.

COMMENT 8)

- Some parts of the Supplementary information (e.g. Section 4) are repetitions of text already present in the Methods section and can be eliminated.

REPLY 8)

We thank reviewer for this observation. We carefully checked the methods and supplementary information and removed repetitions (such as the previous section 4). Changes are marked in red.

Reviewer #2 (Remarks to the Author):

The central claim of the paper by A. Rastelli et. al is the ability to manipulate the direction of the quantization axis of a quantum dot via a piezo-induced stress.

After reading the manuscript, I am convinced that the authors have achieved the claimed effect. The results are new and certainly warrant publication in some form. However, I do not think that this paper meets publication criteria for Nature Communications for two reasons:

We thank the reviewer for the constructive remarks, which we address below.

COMMENT 9)

1) The quantization axis rotation appears to be an "unintended consequence" of frequency-tuning with stress, and it is not predictably controlled by stress. It appears that the degree of orientation control is random. Moreover, it seems that there is no way to predict the behavior of a given dot under stress without an individual characterization of each dot.

REPLY 9)

We first note that **frequency tuning and quantization axis rotation are actually independent effects**. The former depends on the strain through the deformation potentials. The latter depends on the symmetry properties of the system. To illustrate that the axis rotation is not an "unintended consequence" of frequency tuning we note that it would be possible to rotate the quantization axis without changing the emission energy. For bulk GaAs, and using the deformation potential theory, this condition can be achieved by choosing $\varepsilon_{zz} = \varepsilon_{yy} = \varepsilon_{xx} \left(\frac{b-2a}{b+a} \right)$, with a the hydrostatic deformation potential and b one of

the valence band deformation potentials. This configuration corresponds to uniaxial stress along x combined with a biaxial stress in the y - z plane and leads to a well-defined quantization axis along the x -direction (the symmetry in the y - z plane is still cubic) but no shift of the highest lying valence band. Also the opposite is true: one can obtain frequency tuning without changing the quantization axis. This can be easily achieved in practice in a quantum well or in a quantum dot by applying biaxial stress (which is also a more efficient way for frequency tuning compared to the uniaxial stress used in this work) in the growth plane.

The orientation of the new quantization axis is not random - it is deterministically dictated by the orientation of the applied stress. To clearly illustrate this point, we have made the following changes:

- We added a new panel in Fig. 3 (panel d) and also in Fig. S7 (Fig. S7e). These experimental plots clearly show that the transition dipoles align perpendicularly to the new quantization axis (provided by the direction of the applied uniaxial stress), for all measured QDs and thus independent of the peculiarities of each dot.
- On page 8 we have added the sentence: **“However, for sufficiently large stress, the in-plane optical dipoles of X and MX transitions are deterministically aligned perpendicular to the pulling direction, independent of the initial orientation, as shown in Fig. 3d and Fig. S4d and Fig. S7e of the supplementary information.”**

COMMENT 10

2) The authors list potential benefits of quantization axis rotations (see "discussion"), but they are yet to demonstrate any of such benefits experimentally. For instance, if the authors would show any evidence of better (or worse) light-waveguide coupling after axis reorientation, the impact of their work would be significantly stronger.

REPLY 10

We agree with the reviewer. Because of limitations in the available technologies we have not been able yet to implement the concept illustrated in Fig.5, but we are establishing an external cooperation to attempt it. So we hope we'll be able to test the concept soon. We think this is worth the effort, as the coupling efficiency should depend on the relative orientation of the QD transition dipoles \underline{d} with respect to the electric field \underline{E}_k corresponding to the guided mode \underline{k} in a photonic waveguide. The coupling efficiency is in fact proportional to $|\underline{d} \cdot \underline{E}_k|^2$ [Rev. Mod. Phys. 87, 347–400 (2015)]. In addition, also vertically oriented dipoles (usually absent in conventional QDs) are available upon tension. Please refer also to the Reply 1 to Reviewer #1.

COMMENT 11

In addition, there is a minor remark.

It is not obvious to me that fast radiative recombination is beneficial for indistinguishability. I believe that there are several publications on quality of single photons with different confinement, and stronger confinement does not necessarily result in better indistinguishability and photon purity. One of the physical reasons may be a dot re-excitation, see Phys. Rev. Lett. 109, 163601, the other - stronger coupling between acoustic phonons and quantum dots.

REPLY 11

The excitation scheme we have in mind is based on two-photon excitation of XX. In this case recapture processes are strongly suppressed [Appl. Phys. Lett. 112, 093106 (2018) and <https://arxiv.org/abs/1801.01672>], as recently shown for QDs similar to those used in the present manuscript. We have added this remark together with the reference quoted by the Reviewer to explain

the meaning of recapture in the Discussion paragraph: “We stress here that the used QDs have excellent optical properties in terms of single-photon purity and indistinguishability when excited resonantly through a 2-photon-absorption process,^{24,43,44} which is compatible with planar photonic circuits and strongly reduces exciton recapture.^{44,45}”. In addition, we are aware of unpublished data showing that the indistinguishability of subsequently emitted photons from GaAs QDs improves when broadband Purcell enhancement is used. With uniaxial stress we are not affecting substantially the strength of the confinement (the dot size remains practically unaltered and the changes in band-offsets are small). So we are not really moving to a situation of “stronger confinement”, which may be strongly affecting the phonon coupling. We thank nevertheless the Reviewer about this remark. We think it will be interesting to see if there is any measurable effect on the phonon sidebands and to test experimentally the theory predictions.

Reviewer #3 (Remarks to the Author):

This manuscript shows that the quantization axis of GaAs quantum dots can be changed through application of uniaxial strain perpendicular to the growth direction. The quantum dot samples are strained in a piezoelectric actuator that is designed to provide a large, tunable tensile strain, and photoluminescence of single dots is used to observe changes in the emission line strengths, emission energies, and polarization properties that indicate this change in the quantization axis. The results match a theoretical model of the band structure of the semiconductor. The primary motivation for this work is to change the polarization axis of the QD transitions to be more appropriate for in-plane photon emission in photonic structures.

The ability to control the properties of quantum emitters through strain seems quite interesting, and this approach using a piezoelectric device that amplifies strain seems quite novel and provides ways to change the QD properties in ways perhaps not possible before. The idea to change the quantization axis and polarization axes is a novel idea that has potential uses. The paper also combines the experimental observations with theoretical and computation results, making the observations and physics more clear. I recommend it for publication in Nature Communications and think it will be an influential paper in the areas of quantum dots and solid state quantum systems in general.

We thank the Reviewer for recognizing the value of our work and for the insightful comments and questions.

COMMENT 12)

My only criticism is that the language used to describe the wavefunction of electrons in the quantum dot and the quantization axis is confusing to me, and I suspect it may be confusing to others. The main point of confusion for me is what is meant by the quantization axis. When starting to read the paper, I thought it was perhaps referring to the direction in which quantum confinement plays the biggest role – which would ordinarily be the smallest dimension of the quantum dot. After reading more carefully, I came to the conclusion that the paper is about the quantization axis of the total angular momentum for the valence band, which is changed by the application of strain. I’m still not sure that I’m completely understanding this right, so it would be very helpful to clarify this idea.

REPLY 12)

We meant indeed the proper quantization axis for the angular momentum. To make this point clear we, we have extended one sentence in the introduction: “Carriers are therefore strongly confined along the

growth (z) direction, which also defines the natural quantization axis for the total angular momentum operator for the VB states.³⁻⁵

COMMENT 13)

Along these lines, the manuscript discusses how the data in Fig. 1b and 1c show a change in the quantization axis for different strains, but this really isn't clear to me. How is the direction of the quantization axis determined by this data? I'm really not sure what it means to plot the hole character for the x and z quantization axes.

REPLY 13)

We thank the reviewer for this remark. To address it we have done the following changes:

- We explain more precisely what is shown in Figs. 1b and 1c with the following sentence: "To quantify to what extent the quantization axis is oriented along the original z direction or the desired x direction under uniaxial stress, we calculated the HGS of our QDs (Fig. 1a) via the empirical pseudopotential method (EPM) and projected it onto the HH_n , LH_n , and SO_n (split-off) states, i.e. the eigenstates of the angular-momentum-projection operator $J_n = \mathbf{J} \cdot \mathbf{n}$ along the quantization axis direction specified by the unit vector \mathbf{n} . The results for \mathbf{n} parallel to the z- (x-) directions are shown with symbols in Fig. 1b (1c)."
- We provide a more detailed explanation and computational data in Sec. II.1 of the supplementary material and Fig. S9
- We also add the following sentence in the main text: "It is important to note that for intermediate values of strain the overlap of the HGS onto the eigenstates of J_n is rather poor for any \mathbf{n} , indicating that the HGS has low symmetry and a "good" quantization axis cannot be defined. This also mean that uniaxial stress along x produces a quantization-axis flip rather than a smooth rotation. (For more details, see Supplementary Information and provided movie.)"

COMMENT 14)

Related to this are statements about the confinement effects being small compared to the strain effects. That almost sounds like the carriers might not be confined to the dot, but I'm assuming this means that confinement may only play a small role in determining the quantization axis of the angular momentum. I think that clarifying this issue will strengthen the paper significantly.

REPLY 14)

Indeed we mainly meant that at large strains, confinement plays a small role in determining the orientation of the quantization axis. We now state this explicitly in the abstract and text (see Page 4): "Different from most previous experiments, in which stress was added after growth as a perturbation to fine tune the emission properties of QDs,^{25,26} *confinement can be seen here as a perturbation compared to the strain-induced effects in determining the orientation of the quantization axis and hence the symmetry properties of the system.*

I have a few minor questions below that would also be useful to address.

Minor points:

COMMENT 15)

a. On page 9 the manuscript states that there are "two bright excitons (B_y and B_z), which have polarization perpendicular to the new quantization axis x and two dark excitons." When I read this I first

thought it was saying that the dark excitons did not have polarization perpendicular to the quantization axis, but now I think this just means that maybe the electron and hole have a total angular momentum of 2 instead of 1, as in dark excitons with the angular momentum quantized along the z-direction. Could you clarify this point?

REPLY 15)

Indeed this is what we meant. We have added a short explanation: “In the ideal case the Bloch wavefunctions of such dark states (one of which remains dark throughout the experiment) are a linear combination of the states with $J_x = \pm 2$.”

COMMENT 16)

b. Is there any reason why compressive strain could not be applied in this structure to look at the LH_x exciton?

REPLY 16)

Because the membrane thickness (~300 nm) is much smaller than its length (~50 μm), the suspended membrane tends to bend and buckle under compression. We have nevertheless attempted operating the actuator in the compressive regime after manuscript submission. In this case we have used a sample in which the dots are vertically displaced from the neutral plane (see Fig. b below). According to a FEM simulation (see Fig. a below) it should be possible to reach compressive strains of about -0.2% for dots located in an upwards-bent membrane with dots located below the neutral plane (see structure of used sample in b). Fig. c shows grey-scale coded PL spectra of a QD located close to the center of the actuator gap. There is a strong red-shift upon application of a negative electric field to the actuator (tensile regime described in the text) and a weak blue-shift upon application of positive fields (which should correspond to bending-induced uniaxial compressive stress). In theory we expect that under sufficient compressive stress lines should red-shift again, but we were not able to reach this point. If we look at the fine structure of the in-plane-oriented neutral exciton transitions (Fig. f) while moving from the tensile to the compressive regime we see first a rapid decrease down to a minimum (which we identify as the point at which stress is minimal, since QDs in unprocessed samples show a FSS of up to ~10 μeV) and then increases slowly. An example of polarization-resolved spectra is shown in Fig. d for the largest positive field available. The fine-structure (of about 25 μeV) is clearly visible. The high energy component (with intensity higher than the low energy component) is now *aligned parallel to the main stress direction (x, [100] direction)* as expected for an exciton with dominant LH_x character. The polarization direction of the high-energy component of the X emission is shown in Fig. e. We clearly see the expected 90° rotation when moving from a HH_x to a LH_x exciton (see wave-functions in Fig. 1c of the main text).

So, overall, the findings are in line with the predictions. However we prefer not to include these data, since the behavior of the actuator in the compressive regime is not as predictable as in the tensile regime (we cannot control at present the direction of bending, and sometimes the membranes buckle in a complicated manner). We have added the following sentence in the Methods section: “We note that while the actuator shows a fully predictable behavior in the tensile regime, poorly-controlled membrane buckling prevents us to obtain reliable results in the compressive regime.” Since the compressive regime is very interesting (we expect an increased oscillator strength for the dominant line) we are currently exploring alternative strategies (including the application of static stress envisioned in Fig. 5) to achieve it.

REVIEWERS' COMMENTS:

Reviewer #1 (Remarks to the Author):

The authors have properly addressed my concerns and made their claims clearer. The paper can be accepted for publication

Reviewer #3 (Remarks to the Author):

As stated in my report on the original manuscript, I think this is quite interesting work that will be influential in the areas of quantum dots and solid state quantum systems. Being able to change the heavy hole angular momentum quantization axis through strain seems quite impressive and could be quite valuable for in-plane quantum dot photonics. My only real criticisms were that there were a few points that needed clarifying in the paper. The authors have now clarified these points in the paper, and I am satisfied that it is suitable for publication in Nature Communications.